

# Genome sequencing and CAZymes repertoire analysis of *Diaporthe eres* P3-1W causing postharvest fruit rot of 'Hongyang' kiwifruit in China

Li-Zhen Ling[1], Ling-Ling Chen[1,2], Zhen-Zhen Liu[1], Lan-Ying Luo[1], Si-Han Tai[1] and Shu-Dong Zhang[1]

[1] School of Biological Sciences and Technology, Liupanshui Normal University, Liupanshui, Guizhou, China
[2] College of Life and Health, Dalian University, Dalian, Liaoning, China

Corresponding author
Shu-Dong Zhang,
sdchang@foxmail.com

## ABSTRACT

Postharvest rot caused by various fungal pathogens is a damaging disease affecting kiwifruit production and quality, resulting in significant annual economic losses. This study focused on isolating the strain P3-1W, identified as *Diaporthe eres*, as the causal agent of 'Hongyang' postharvest rot disease in China. The investigation highlighted cell wall degrading enzymes (CWDEs) as crucial pathogenic factors. Specially, the enzymatic activities of cellulase, β-galactosidase, polygalacturonase, and pectin methylesterases peaked significantly on the second day after infection of *D. eres* P3-1W. To gain a comprehensive understanding of these CWDEs, the genome of this strain was sequenced using PacBio and Illumina sequencing technologies. The analysis revealed that the genome of *D. eres* P3-1W spans 58,489,835 bp, with an N50 of 5,939,879 bp and a GC content of 50.7%. A total of 15,407 total protein-coding genes (PCGs) were predicted and functionally annotated. Notably, 857 carbohydrate-active enzymes (CAZymes) were identified in *D. eres* P3-1W, with 521 CWDEs consisting of 374 glycoside hydrolases (GHs), 108 carbohydrate esterase (CEs) and 91 polysaccharide lyases (PLs). Additionally, 221 auxiliary activities (AAs), 91 glycosyltransferases (GTs), and 108 carbohydrate binding modules (CBMs) were detected. These findings offer valuable insights into the CAZymes of *D. eres* P3-1W.

# INTRODUCTION

Kiwifruit, an economically significant fruit crop in the *Actinidia* genus of the Actinidiaceae family, has its center of diversity in China. To date, China have recorded approximately 52 identified *Actinidia* species (*Huang, 2016*). The history of kiwifruit cultivation dates back to the early 20[th] century when wild seeds from China were introduced to New Zealand (*Ferguson, 1984*). Over time, a variety of cultivation varieties have been developed, with *A. chinensis* and *A. chinensis* var. *deliciosa* emerging as the two most commercially important kiwifruit varieties (*Waswa et al., 2024*). The 'Hongyang' kiwifruit (*A. chinensis*)

stands out as the first cultivar with a red-flesh inner pericarp, originating from clonally selected wild germplasm in central China (*Zhen et al., 2004*). 'Hongyang' has garnered significant attention over the past four decades and is extensively cultivated in China due to its delectable taste and rich nutritional profile, including high levels of vitamin C, minerals, carotenoids, and anthocyanins (*Wang, Qiu & Zhu, 2021a*).

'Hongyang' kiwifruit is highly susceptible to soft rot disease both during cultivation and postharvest storage, lead to significant annual economic losses (*Lei et al., 2019*; *Ling et al., 2023*). Researches have indicated that various fungal pathogens from the genera *Botrytis*, *Diaporthe*, and *Alternaria* are the culprits behind fruit rot during storage (*Li et al., 2017a*; *Ling et al., 2023*). Among these pathogens, *Botrytis* spp. and *Diaporthe* spp. are widely acknowledged as the most severe threats to stored kiwifruit (*Díaz et al., 2017*; *Zhou et al., 2015*). The primary method currently used to combat these pathogens involves the environmentally harmful fungicides, which can leave residues in the fruit and contribute to the development of antifungal resistance (*Bardas et al., 2010*). An alternative approach includes the use of biocontrol agents to target multiple postharvest pathogens (*Francesco, Martini & Mari, 2016*). The effectiveness of these pathogen control strategies is mainly due to various action mechanisms, such as the production of antifungal compounds and cell wall degrading enzymes (CWDEs). To minimize the impact of this destructive disease, a deeper understanding of the pathogenicity mechanisms of the responsible species is crucial.

The complete genome sequences of the causal agents provide crucial information for studying their pathogenicity. Currently, there is a wide variety of different next-generation sequencing (NGS) technologies available (*Di Bella et al., 2013*). For example, Illumina's HiSeq can generate numerous short reads with high sequencing depth at relatively low costs, with mean error rates <1% (*Laehnemann, Borkhardt & McHardy, 2016*). However, challenge of assembly must be addressed as short reads often contain complex repeats. By utilizing the third-generation sequencing platform Pacific Bioscience (PacBio) with single-molecule real-time (SMRT) sequencing technology, long reads exceeding 20 kb can be produced, overcoming assembly difficulties associated with short reads. Although this platform has been reported to have lower sequencing depth and higher error rates (*Laehnemann, Borkhardt & McHardy, 2016*), combining PacBio and HiSeq reads can improve assembly contiguity and accuracy per base (*Laehnemann, Borkhardt & McHardy, 2016*).

Plant cell walls serve as one of the primary defense mechanisms against pathogen invasion and are predominantly composed of polysaccharides such as hemicellulose, cellulose and pectin (*Chen et al., 2018*). Pathogens secrete carbohydrate active enzymes (CAZymes) to break down these cell wall polysaccharides (*Castillo et al., 2017*). The CAZymes are categorized into six classes by the carbohydrate-active enzyme (CAZy) database: auxiliary activity (AA), carbohydrate esterase (CE), carbohydrate-binding modules (CBMs), glycoside hydrolase (GH), glycosyl transferase (GT), and polysaccharide lyases (PL) (*Drula et al., 2022*). Recent genome sequencing efforts have identified numerous CAZymes candidates across various pathogenic species (*Castillo et al., 2017*). Fungal pathogens have been shown to rely on cell wall degrading enzymes (CWDEs) for

their pathogenicity. A range of CWDEs, including cellulase (Cx), β-galactosidase (β-Gal), polygalacturonase (PG) and pectin methylesterases (PME), facilitate the breakdown of cell wall polysaccharides, promoting successful pathogen infection (*Ramos et al., 2016a*). Alterations in CWDE activities have been linked to disease progression in plants like apples (*Miedes & Lorences, 2006*), grapefruits (*Shi et al., 2019*) and pumpkins (*Li et al., 2023*).

In this study, a pathogenic strain P3-1W was isolated from diseased 'Hongyang' kiwifruit and identified using morphological and molecular data. The activities of four CWDEs, including Cx, β-Gal, PG and PME were investigated at different stages of P3-1W infection. The genome of P3-1W was sequenced and assembled using Illumina NovaSeq and PacBio SMRT sequencing technologies. Subsequently, a genome-wide identification of genes-encoding CAZymes were performed. Therefore, the combination of genome description and enzyme assay results will enhance our understanding of the mechanisms underlying the pathogenicity of the strain P3-1W.

## MATERIALS AND METHODS

### Pathogen isolation and pathogenicity test

Decayed 'Hongyang' fruits were collected from a cold storage facility at a fruit market in Liupanshui City, Guizhou Province. The pathogen was isolated and purified from the diseased fruit using a standard tissue isolation method as detailed in a previous study (*Ling et al., 2023*). To assess the pathogenic potential of 'Hongyang' soft rot disease, mycelial plugs (5 mm in diameter) of the pathogen were transferred from potato dextrose agar (PDA) media onto the surface of fresh healthy kiwifruits. These fruits were then incubated at 25 °C in the darkness until the symptomatic tissues were appeared. The pathogen was subsequently re-isolated from the diseased tissues, With sterile PDA plugs of the same size serving as control. Three trials were conducted, each consisting of five fruits.

### Isolation and identification of pathogens

The pathogenic strain was identified using a combination of morphological and molecular methods. Morphological identification involved culturing the strain on PDA media at 25 °C for 3 days. For molecular identification, pathogen DNA was extracted using the fungal DNA isolation kit from Sangon Biotech, Shanghai, China, following the manufacturer's protocol. The internal transcribed spacer (ITS) and translation elongation factor 1 (*TEF1*) gene segments were amplified using common primers described in Table S1. The PCR system consisted of 12.5 μL 2 × Taq PCR mix, 0.4 μL of each primer (10 μmol $L^{-1}$), 1 μL DNA template, and 8.2 μL double distilled water. The PCR reaction followed a program with initial denaturation at 95 °C for 3 min, 35 cycles of denaturation at 95 °C for 20 s, annealing at 58 °C for 30 s, extension at 72 °C for 1 min and a final extension at 72 °C for 5 min. The resulting PCR product was sequenced by Sangon Biotech, Shanghai, China.

ITS (PP256503) and *TEF1* segments (PP265422.1) were used for phylogenetic analysis. The corresponding sequences from 86 *Diaporthe* species were downloaded from the NCBI database (Table S2). The sequences of two loci were concatenated to create a single

alignment dataset for phylogenetic inference using maximum likelihood analysis (ML). The phylogenetic tree was constructed using RAxML v7.2.6 (*Stamatakis, 2006*) under the GTR + gamma model with 1,000 bootstraps (*Drummond et al., 2012*).

## Crude enzyme extraction and assay of enzymic activity

The strain P3-1W was cultured on PDA. Mycelial plugs (5 mm) were cut from the edge of a 3-day-old colony of the strain and transferred to the surface of 'Hongyang' fruit with four or five pinholes. A blank PDA plug of the same size was used as the negative control group. The inoculated fruits were sealed in a vessel and incubated in dark conditions, with disease progression monitored. The lesion margins between infected and healthy fruit were collected daily for examining enzyme activity.

Crude enzyme extraction was performed following Chen's method (*Chen et al., 2018*) with some modifications. Briefly, 3 g of fresh fruit tissue was homogenized with 12 mL of 2 M NaCl buffer solution (containing 10 mmol/L EDTA and 5 g/L PVP) adjusted to pH 7.4 using 0.01 mol $L^{-1}$ NaOH at 4 °C. The homogenate was then centrifuged at 15,000 × $g$ for 30 min at 4 °C, and the resulting supernatant was used as the crude enzymatic extract.

The enzyme activities of cellulase (Cx), PG and PME were determined using 3,5-dinitrosalicylic acid (DNS) colorimetric method. The reaction mixture consisted of 1.0 ml of substrate solution, 1.0 ml of sodium acetate buffer (pH 4.4) and 0.5 ml crude enzyme, with the reaction initiated by adding the enzyme and incubated at 37 °C for 30 min. A volume of 1.5 mL of DNS was then added to the reaction mixture after the reaction was terminated by boiling for 5 min and the OD values of the reducing production were measured at 540 nm. The blank consisted of the reaction mixture with boiled crude enzyme. Each experiment was repeated three times. The substrates for Cx, PG, and PME were 1% (w/w) carboxymethyl cellulose, 1.0% (m/v) polygalacturonic acid, and 1.0% (m/v) pectin, respectively. The activities of PG and PME were expressed as reducing units (RU), with one RU defined as the amount of enzyme needed to release reducing groups at 1 μmol/min using D-galacturonic acid as a standard. In contrast, one unit (U) of cellulase activity was defined as the micromoles of glucose released per minute of reaction using glucose as a standard.

β-Galactosidase (β-Gal) activity was measured in a reaction mixture containing 5.0 mL of 20 mmol sodium acetate (pH 4.7), 2 mL of 3 mmol/L p-nitrophenyl-β–D-galactopyranoside, and 1.0 mL of crude enzyme. The reaction was carried out at 37 °C for 30 min. Subsequently, 2 mL of 0.2 mmol/L $Na_2CO_3$ was added to halt the reaction. The concentration of the reducing product was quantified at 420 nm using p-nitrophenol (PNP) as a standard. One unit of β-Gal activity was defined as the enzyme amount that generated 1 mmol of PNP per hour.

## Genomic DNA extraction and sequencing

Up to 100 mg of pathogen mycelia were collected from Petri dishes and frozen with liquid nitrogen. Genomic DNA of the pathogen was extracted using DNeasy Plant Kit (Qiagen, Hilden, Germany) following the manufacturer's protocol. The quality of DNA was assessed using Nanodrop 2000c (Thermo Scientific, Waltham, MA, USA).

For Illumina sequencing platform, the 350 bp paired-end libraries were constructed following to the manufacturer's instructions and then sequenced on the Illumina Novaseq 6000 platform by Berry Genomics Company in Beijing, China. The library preparation involved DNA fragmentation by sonication, end-polishing, A-tailing ligation with Illumina adapters, PCR amplification, and purification using AMPure XP bead system. Size distribution of the libraries was analyzed using an Agilent 2100 Bioanalyzer.

For PacBio SMRT bell library preparation, 7 μg of high-quality genomic DNA was evaluated for size using pulsed-field electrophoresis, ensuring most fragments were longer than 20 Kb. DNA was sheared into a mode size of 40 Kb or larger using g-TUBE and then concentrated using AMPure® PB Beads. The SMRTbell library was prepared using Kit 2.0, involving removal of single-strand overhangs, DNA damage repair, end-repair, A-tailing, adapter ligation and enzymatic digestion. Library size-selection was performed using SageELF. Finally, the library was sequenced for 15/30 h on the Sequel II/IIe system (Pacific Biosciences, Menlo Park, CA, USA).

## Genome assembly and annotation of P3-1W

Illumina sequencing reads were initially utilized to estimate genome size and heterozygosity through Jellyfish (*Hesse, 2023*) and GenomeScope (*Ranallo-Benavidez, Jaron & Schatz, 2020*). Following the removal of the low-quality reads, the resulting clean reads were employed for *de novo* assembly into contigs and scaffolds using SOAPdenovo software (*Bankevich et al., 2012*). The quality of the genome assembly was evaluated using Benchmarking Universal Single-Copy Orthologues (BUSCO) (*Simão et al., 2015*). Subsequently, the assembled genome was subjected to RepeatMasker (*Tarailo-Graovac & Chen, 2009*) to mask repeat sequences and annotate transposable elements (TEs). Prediction of protein-coding genes (PCGs) was carried out with Funannotate (https://GitHub.com/nextgenusfs/funannotate). Identification of tRNA and rRNA genes was conducted using tRNAscan-SE and BAsic Rapid Ribosomal RNA Predictor (barrnap), respectively. snRNAs were annotated using Rfam with default parameters. Functional annotation of PCGs was accomplished by performing BLASTP searches against NR, COG, GO, and KEGG databases.

## Identification and comparative analysis of CAZymes

Carbohydrate-active enzyme (CAZyme) searches were conducted using HMMER 3.0 package against Pfam Hidden Markov Models (HMMs) available from dbCAN database (*Zheng et al., 2023*). DIAMOND was utilized for blast hits in the CAZy database.

Comparative analyses included eight additional fungal genomes: *D. amygdali* CAA958, *D. eres* CBS 160.32, *D. capsici*, *D. citri* ZJUD2, *D. citriasiana* ZJUD30, *P. longicolla*, *D. batatatis*, and *D. phragmitis* NJD1. Among them, *D. phragmitis* NJD1 was identified as a causal agent of kiwifruit rot (*Wang et al., 2021b*) and was used for genome characteristic comparison. The abundance of CAZymes was compared across seven other genomes, which primarily consisted of pathogens of soybean (*Li et al., 2017b*), citrus (*Gai et al., 2021*), grapevine (*Morales-Cruz et al., 2015*), blueberry (*Hilário et al., 2022*), sunflower

(*Baroncelli et al., 2016*), walnut (*Fang et al., 2020*) and sweet potato (*Yang et al., 2022*) in this study.

## Statistical analysis

Analysis of variance (ANOVA) was used to analyze the data with SPSS version 17.0 software. Student's t-test was utilized to compare the mean values of the dataset. A *P*-value of less than or equal to 0.05 or 0.01 was considered statistically significant.

## RESULTS

### Isolation and identification of pathogens of 'Hongyang' soft rot disease

In this study, a total of 11 isolates were obtained and initially identified using the internal transcribed spacer (ITS) region. BLAST analysis revealed that these isolates belonged to three fungal taxa: *Diaporthe* spp. (eight isolates), *Alternaria* spp. (two isolates) and *Phomopsis* spp. (one isolate) (Table S3). Among them, the fungal isolate P3-1W, representing *Diaporthe* spp., was selected for further analysis in this study. Following incubation, the colony of P3-1W exhibited a round and cream-like appearance on PDA media, with a white surface and brown underside (Figs. 1A and 1B). Pathogenicity tests were conducted on the healthy 'Hongyang' fruits, where 4 to 5 needles were slightly wounded on the epidermis per fruit were and inoculated with 5-mm mycelial plugs from P3-1W. A sterile PDA plug was severed as the negative control treatment. The obvious symptoms of soft rot appeared 5 days after post-inoculation, while the control fruits remained asymptomatic (Figs. 1C and 1D). Furthermore, P3-1W was successfully reisolated from the inoculated 'Hongyang' fruits, exhibiting consistent morphological characteristics with the original strain. Phylogenetic analysis based on ITS and translation elongation factor 1 (*TEF1*) segments placed the strain P3-1W in a clade with *D. eres* (Fig. S1), confirming its identification as *D. eres*. Additionally, evident rot symptoms were observed when mycelial plugs from P3-1W were inoculated into the epidermis of other fruits, such as cherry tomatoes, black fructus and kumquats (Fig. 1E). These findings indicated that the P3-1W isolate can induce a broad spectrum of fruit rot diseases in kiwifruit and other plant species.

### The activity changes of CWDEs during infection of *D. eres* P3-1W

Previous studies have demonstrated that pathogens can release CWDEs as pathogenicity factors to disrupt the plant cell wall barrier (*Plaza, Silva-Moreno & Castillo, 2020*; *Quoc & Chau, 2017*). Some CWDEs such as polygalacturonase (PG), pectin methylesterase (PME), pectate lyase (PL), β-galactosidase (β-Gal) and cellulase (Cx) have been found to be activated during pathogen infection (*Chen et al., 2018*). In our study, we investigated the activities of β-Gal, Cx, PG and PME at the different stages of *D. eres* P3-1W infection on 'Hongyang' kiwifruit, with raw data provided in Table S4. Our results showed that all enzymes except β-Gal exhibited similar trends during *D. eres* P3-1W infection (Fig. 2). The activities of these three enzymes began to rise after infection, pecked at the second day post-infection (dpi), decreased from 3 dpi, and then increased again at 5 dpi (Figs. 2A–2C). These enzyme activities during *D. eres* P3-1W infection were significantly higher than the

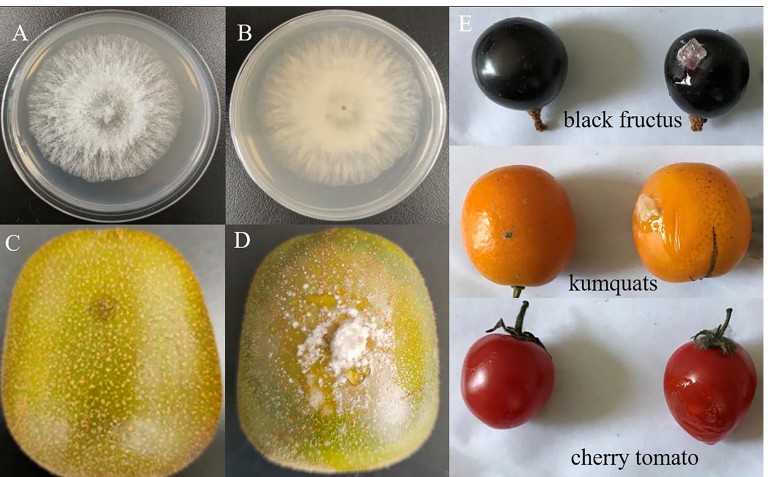

**Figure 1 The colony morphology of strain P3-1W isolated from diseased 'Hongyang' fruits and its pathogenicity test.** The colony morphology of strain P3-1W isolated from diseased 'Hongyang' fruits: (A) the front of colony and (B) the back of colony and symptoms of soft rot in 'Hongyang' fruit artificially inoculated mycelial plugs of P3-1W (C) and control (D) for 7 days after inoculation, and (E) other three fruits for 5 days after inoculation.    

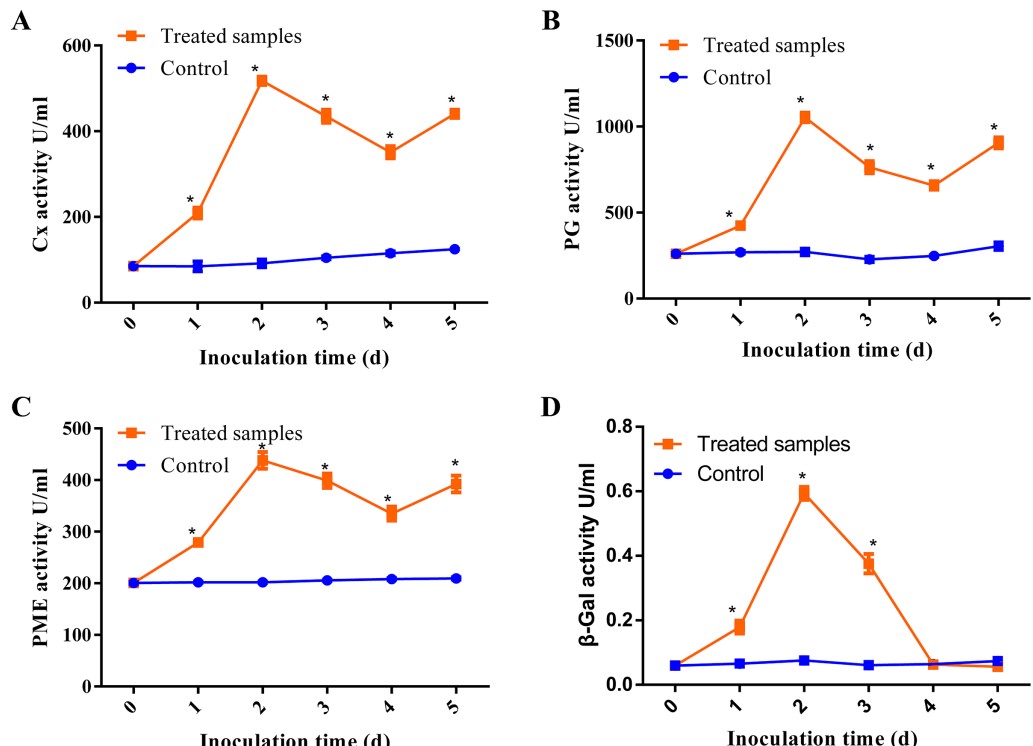

**Figure 2 The activities of CWDEs during the different times of *D. eres* P3-1W infection.** Activity of (A) Cx; (B) PG; (C) PME and (D) β-Gal. Values are expressed as mean values and standard errors. (*$P < 0.05$).    
| Table 1 Genome sequencing statistics of *D. eres* P3-1W. | | |
|---|---|---|
| Illumina | Total reads | 97,013,864 |
| | Coverage | 242.15× |
| | Raw data | 14,552,079,600 bp |
| | Clean data | 13,901,222,433 bp |
| | Estimated size | 58.9 Mb |
| | Heterozygosity | 17.5% |
| PacBio | Subreads bases | 12,243,480,442 bp |
| | Coverage | 208.78× |
| | Subreads reads | 613,696 |
| | Subreads mean length | 19,950 bp |
| | Subreads N50 | 22,159 bp |

control at the same infection time. The activity of β-Gal showed minimal variation compared to the other enzymes. Its activity increased significantly in the first 2 days after P3-1W infection, then sharply declined from 3 dpi, reaching control levels at the fifth dpi (Fig. 2D). These findings indicated that *D. eres* P3-1W can secret CWDEs and modulate enzymatic activities during infection on 'Hongyang' kiwifruit.

## The characteristics and annotation of *D. eres* P3-1W genome

In this study, a hybrid assembly of two sequencing platforms was utilized to obtain the genome of *D. eres* P3-1W for a comprehensive identification and characterization of CWDEs. The Illumina platform generated 97,013,864 paired-end short reads (~242 ×) (Table 1), which underwent quality control and adapter trimming. The resulting clean reads were used to analyze the genome characteristics of *D. eres* P3-1W. Kmer analysis estimated the genome size at 58.9 Mb, with a heterozygosity ratio of 17.5% (Table 1). In addition, PacBio long reads (~20 kb) sequencing generated approximately 613,696 subreads (about 12 Gb, ~208 ×) with mean and N50 subread lengths of 19,950 and 22,159 bp, respectively (Table 1). The *de novo* assembly of the *D. eres* P3-1W genome was conducted using SOAPdenovo software, resulting in 14 scaffolds. The assembled genome size was 58,489,835 bp, with an N50 of 5,939,879 bp and a GC content of 50.7% (Table 2). Furthermore, the completeness of the genome assembly using BUSCO, revealed that 97.6% of the conserved core gene sets were present (Table 2). By contrast, *D. phragmitis* NJD1 was found to cause the kiwifruit soft rot and its genome sequenced by Illumina and PacBio has been released (*Wang et al., 2021b*). However, its genome assembly consisted of 28 contigs with a contig N50 of 3,550,333 bp (Table 2). Apparently, the genome assembly of *D. eres* P3-1W in this study exhibited a higher completeness compared to that of *D. phragmitis* NJD1.

A total of 1,473,598 bp of the repetitive sequences were identified using RepeatMasker (*Tarailo-Graovac & Chen, 2009*), representing 2.52% of the *D. eres* P3-1W genome

**Table 2 The genome features of *D. eres* P3-1W compared with *D. phragmitis* NJD1.**

|  | *D. eres* P3-1W | *D. phragmitis* NJD1 |
|---|---|---|
| Genome size (bp) | 58,489,835 | 58,328,132 |
| GC% | 50.7 | 50.82 |
| N50 (bp) | 5,939,879 | 3,550,333 |
| Number of scaffolds/contigs | 14 | 28 |
| Repeat sequence (bp) | 1,473,598 | / |
| Total genes | 15,618 | 12,393 |
| PCGs | 15,407 | 12,393 |
| ncRNAs | 23 | 16 |
| rRNAs | 45 | 37 |
| tRNAs | 143 | 174 |
| CAZymes | 857 | 806 |
| Genome BUSCO% | 97.64 | 97.90 |
| NR annotation | 14,988 | 11,624 |
| COG annotation | 6,003 | 2,206 |
| GO annotation | 7,656 | 7,853 |
| KEGG | 2,979 | 10,207 |

(Table 2). The masked genome sequence was used for the *ab initio* gene prediction, resulting in the identification of 15,407 protein-coding genes (PCGs) with an average length of 1,584 bp in *D. eres* P3-1W genome (Table 2). In addition, 143 tRNA genes, 45 rRNA genes, and 23 non-coding RNA genes were also identified in the genome of *D. eres* P3-1W (Table 2). Functional annotation of these PCGs revealed that 97.28% out of them were annotated in various databases (Table 2), with 14,988 genes showing sequence similarity to orthologous proteins in the NCBI-NR database. Furthermore, 6,003 PCGs were assigned to different COG categories (Fig. 3), with the most abundant category being 'Carbohydrate metabolism and transport', followed by 'Secondary Structure'. Among the three GO classifications, the most common molecular functions of PCGs were 'binding' and 'catalytic activity', while the majority of biological processes were associated with 'cellular processes' and 'metabolic processes'. The three most abundant cellular components were 'cell', 'organelle' and 'membranes' (Fig. S2). KEGG pathway analysis of these PCGs revealed that 2,979 genes were annotated in the KEGG database (Table 2), with 'global view and maps' being the most enriched term, followed by 'amino acid metabolism' and 'carbohydrate metabolism' (Fig. S3). While the PCGs in the *D. eres* P3-1W genome shared functional similarities with those in *D. phragmitis* NJD1, differences were observed in the COG and KEGG pathway distributions (Table 2).

### Genome-wide identification of CAZyme genes in *D. eres* P3-1W

In this study, a total of 857 genes encoding putative CAZymes were identified, representing 5.49% of the predicted genes (Table 2 and Fig. 4) in *D. eres* P3-1W. Statistical analysis revealed significant differences in the number of CAZymes among eight *Diaporthe* species
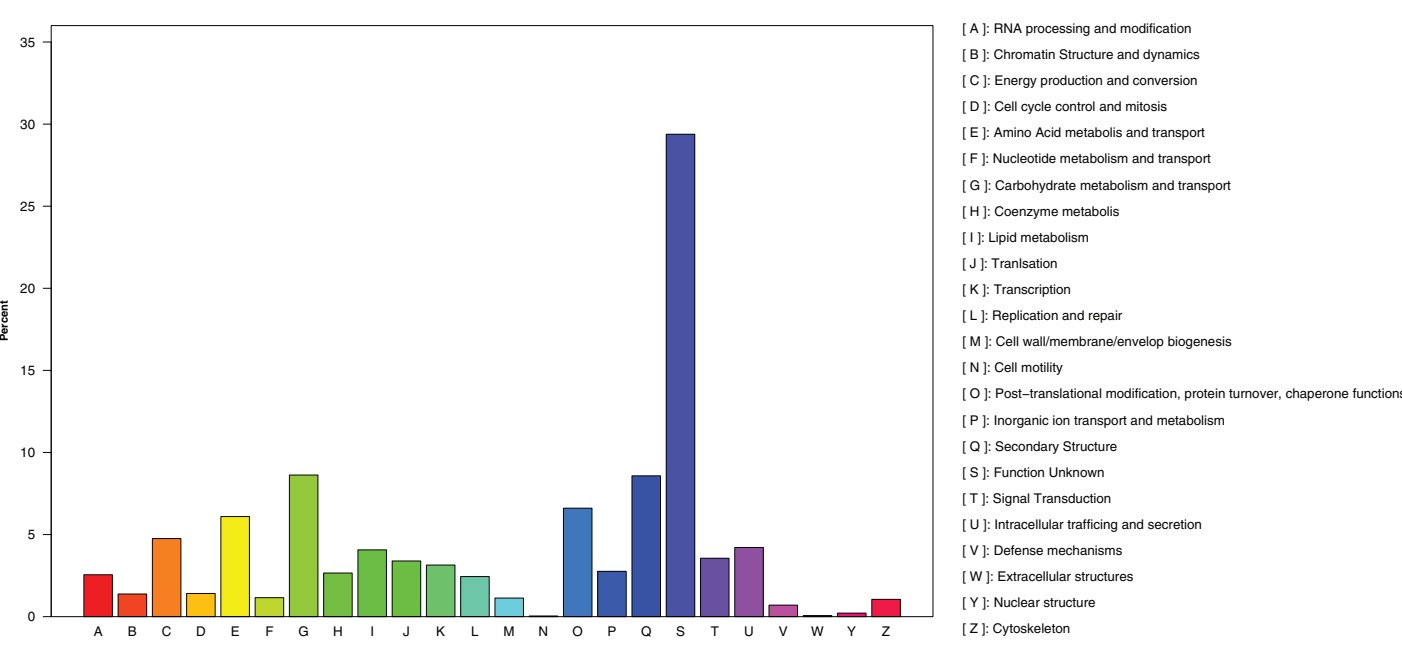

**COG Function Classification**

[ A ]: RNA processing and modification
[ B ]: Chromatin Structure and dynamics
[ C ]: Energy production and conversion
[ D ]: Cell cycle control and mitosis
[ E ]: Amino Acid metabolis and transport
[ F ]: Nucleotide metabolism and transport
[ G ]: Carbohydrate metabolism and transport
[ H ]: Coenzyme metabolis
[ I ]: Lipid metabolism
[ J ]: Tranlsation
[ K ]: Transcription
[ L ]: Replication and repair
[ M ]: Cell wall/membrane/envelop biogenesis
[ N ]: Cell motility
[ O ]: Post–translational modification, protein turnover, chaperone functions
[ P ]: Inorganic ion transport and metabolism
[ Q ]: Secondary Structure
[ S ]: Function Unknown
[ T ]: Signal Transduction
[ U ]: Intracellular trafficing and secretion
[ V ]: Defense mechanisms
[ W ]: Extracellular structures
[ Y ]: Nuclear structure
[ Z ]: Cytoskeleton

**Figure 3 COG categories assigned to the PCGs of *D. eres* P3-1W.**

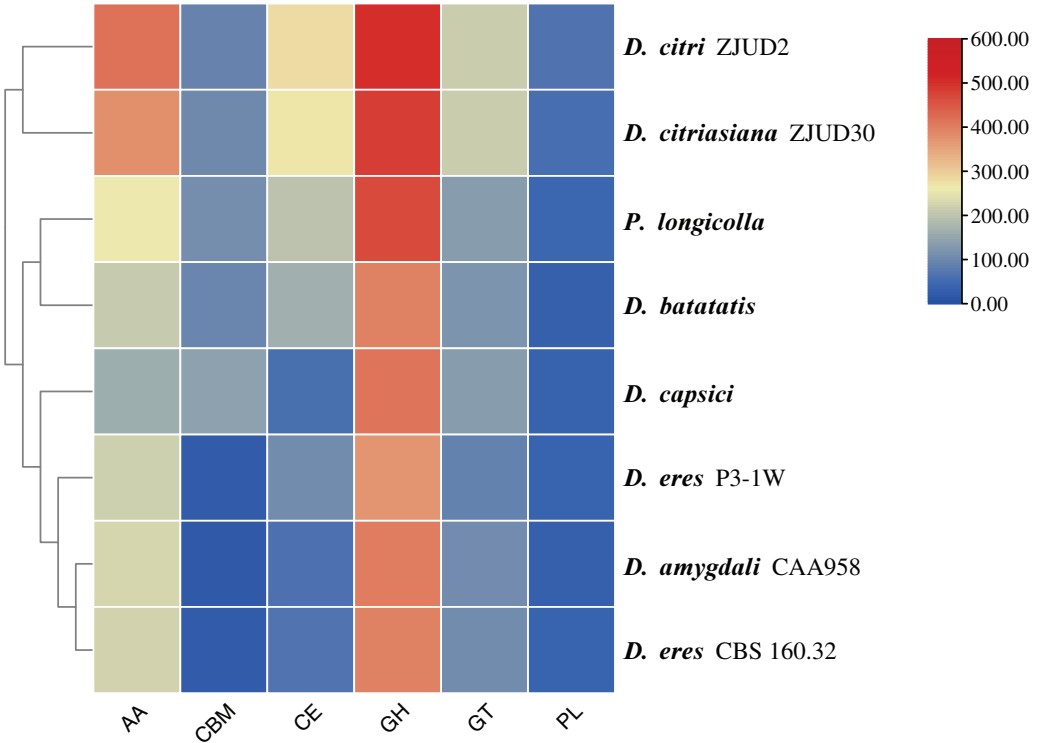

**Figure 4 Cluster analysis of CAZymes in *D. eres* P3-1W and seven other *Diaporthe* species.**

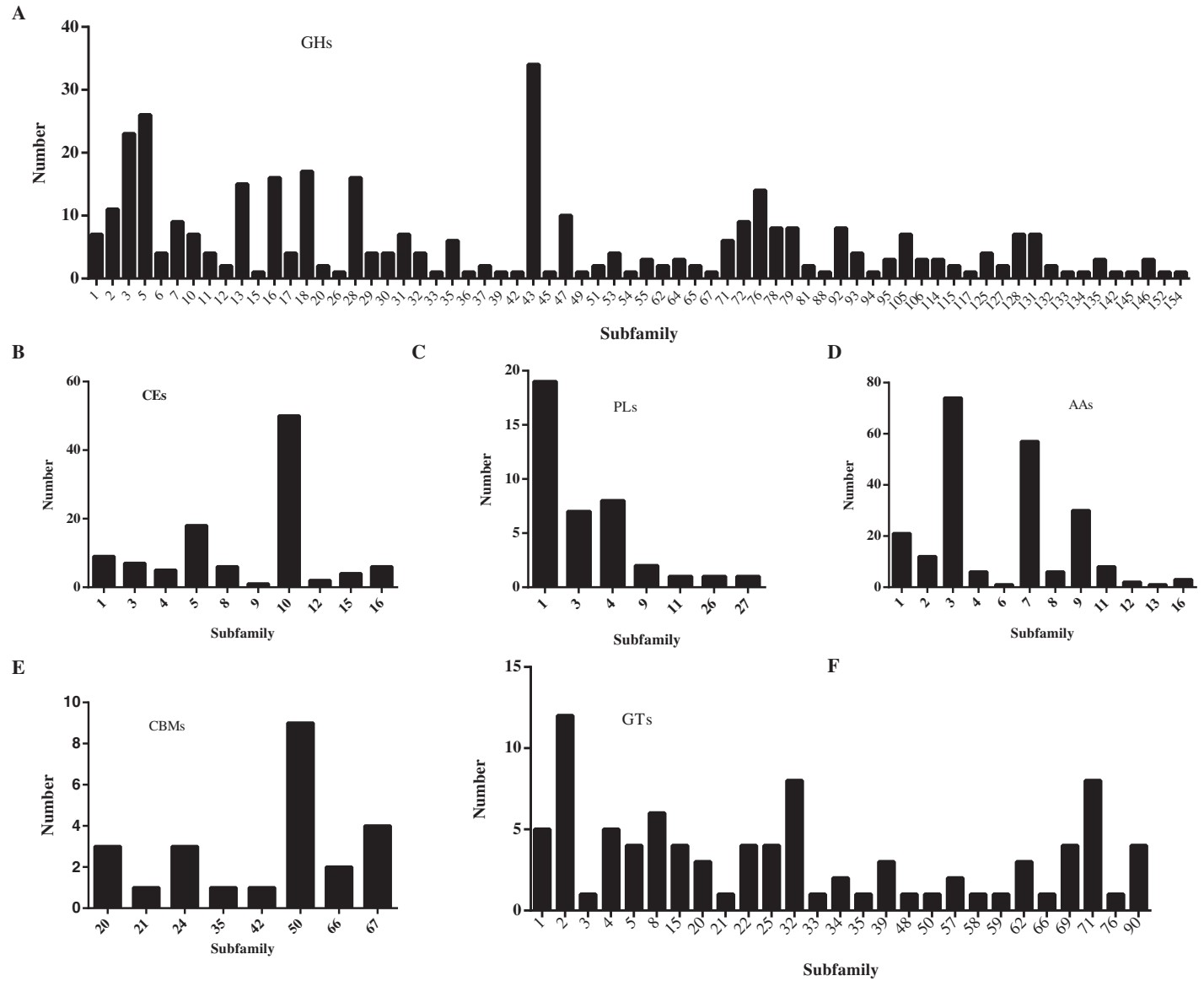

**Figure 5 Number of CAZymes in *D. eres* P3-1W genome.** Number of (A) GHs subfamilies; (B) CMBs subfamilies; (C) PLs subfamilies; (D) AAs subfamilies; (E) CEs subfamilies and (F) GTs subfamilies.

($p < 0.05$), which were further classified into three clusters. *D. eres* P3-1W formed one cluster with three other species (Fig. 4). Comparative analysis of CWDEs in *D. eres* P3-1W was conducted with seven other fungi species (Fig. 4). The results demonstrated that CWDEs comprising of glycoside hydrolase (GH), carbohydrate esterase (CE) and polysaccharide lyase (PL) were present in each species (Fig. 4). The total number of CWDEs ranged from 500 in *D. amygdali* to 851 in *D. citri* ZJUD2 (Fig. 4). *D. citri* ZJUD2 possessed the highest amount of GH, CE, and PL genes with 505, 281 and 65 copies, respectively (Fig. 4), while the lowest number of GHs, CEs and PLs was appeared in different species. For example, *D. eres* P3-1W had the lowest number of GHs with 374 copies. Common CWDEs for plant cell wall degradation, such as cellulase (GH3, −5, −6, −7, −12, and −45),

pectinase (GH28) and hemicellulose (GH11 and GH43) were found in *D. eres* P3-1W (Fig. 5A) were observed in *D. eres* P3-1W. Additionally, genes related to xylanase (GH10, −11, and −30) and chitinase (GH18) were also found in *D. eres* P3-1W (Fig. 5A). Additionally, this species harbored a diverse array of polysaccharide-deacetylating CE enzymes with 108 copies, compared to *D. capsica* with 60 copies (Figs. 4 and 5B). *D. amygdali* CAA958 and *D. batatatis* had the smallest number of PLs with 33 copies each (Fig. 5C), while *D. eres* P3-1W had 39 copies belonging to 7 subfamilies (Fig. 5C), with pectase lyases PL1 being the most prominent subfamily (Fig. 5C).

In addition to CWDEs, the genomes of these species also contained other CAZymes such as auxiliary activity (AA), carbohydrate binding module (CBM) and glycosyltransferase (GT), which play crucial roles in lignin depolymerization and carbohydrate utilization from host plants. *D. eres* P3-1W was found to a total of 336 copies of CAZymes, the smallest number among the eight species (Fig. 4). Specifically, 221 AAs, 24 CBMs and 91 GTs were identified in *D. eres* P3-1W. Our findings showed that *D. eres* P3-1W possessed 12 AA subfamilies (Fig. 5D), with cellobiose dehydrogenases (AA3) subfamily being the most abundant (74 copies) and the xylo- and cello-oligosaccharide oxidases (AA7) subfamily coming in second with 57 copies (Fig. 5D). In addition, 30 copies of lytic polysaccharide monooxygenases (LPMOs) from the AA9 subfamily were detected in *D. eres* P3-1W (Fig. 5D). The CBMs were also identified, which can form a two-domain structure together with catalytic domains (CDs) of cellulases by increasing the enzyme concentration on the substrate surfaces. Notably, CBM50 (nine copies) was the most abundant subfamily, followed by CBM 67 (four copies) in *D. eres* P3-1W (Fig. 5E). Within the GT family, GT32 and GT71 subfamilies were the most abundant (Fig. 5F).

## DISCUSSION

In this study, *Diaporthe* strain P3-1W was identified as the causal agent of 'Hongyang' postharvest rot disease. In the early years, one ITS segment was sufficient to identify *Diaporthe* species (*Mathew et al., 2015*; *Ménard et al., 2014*), the evolving taxonomy now requries at least two loci segments (ITS and *TEF1*) for accurate identification (*Santos, Alves & Alves, 2017*). Our results confirmed that strain P3-1W was *D. eres* based on ITS and *TEF1* segments. Previous studies have linked various *Diaporthe* species, such as *D. phragmitis* (*Wang et al., 2021b*), *Phomopsis longicolla* (*Liu et al., 2020*), *D. perniciosa*, *D. actinidiae* (*Lee et al., 2001*), *D. ambigua* (*Auger, Pérez & Esterio, 2013*), *D. novem* (*Díaz et al., 2014*), *D. australafricana*, *D. rudis* (*Díaz et al., 2017*), *D. lithocarpus* (*Li et al., 2016*) and *D. viticola* (*Luongo et al., 2011*) to kiwifruit decay. *D. eres* has been recently associated with fruit rot in hardy kiwifruit (*A. arguta*) in China (*Liu et al., 2021a*) and *A. deliciosa* kiwifruit in South Korea (*Park, Kim & Yang, 2023*) To our knowledge, this study marks the first report of postharvest rot disease caused by *D. eres* on 'Hongyang' kiwifruit in China. Additionally, *D. eres* has been found to cause fruit rot in various economically important plants, including yellow peach (*Xiao et al., 2022*), European pear (*Bertetti et al., 2018*), persimmon (*Geng & Wang, 2023*), cherry tomato, black fructus and kumquats as well in

this study. These findings suggested *D. eres* should have broad host range. Moreover, studies have indicated that latent *Botrytis* colonization in healthy flowers and floral parts in fruit orchids can cause disease occurrence during kiwifruit storage (*Riquelme-Toledo et al., 2020*; *Riquelme et al., 2021*). While *Diaporthe* species have been identified 'Hongyang' kiwifruit orchids (*Lei et al., 2019*; *Hui et al., 2018*). Further research is need to determine their roles in postharvest rot during storage.

Cell wall degrading enzymes (CWDEs) secreted by pathogens play a cruial role in the degradation of plant cell walls during pathogenesis (*Chen et al., 2018*). For instance, *Ramos et al. (2016b)* discovered that *Macrophomina phaseolina* secretes polygalacturonase (PG), pectin methylgalacturonase (PMG), and cellulase (Cx), each exhibiting distinct activities in the degradation of maize and sunflower cell walls. Our study also revealed that PG and pectin methylesterase (PME) exhibited higher activities compared to β-galactosidase (β-Gal) and Cx during *D. eres* P3-1W infection of 'Hongyang' kiwifruit. A variety of CWDEs, such as PG, pectinlyase (PL), pectate lyase (PNL), PMG, pectinesterase (PE), Cx, β-glucosidase, and xylanases are present in pathogenic fungi (*Tingley et al., 2021*). Therefore, obtaining the genome sequence of *D. eres* P3-1W is essential to comprehensive understanding of these CWDEs. Next-generation sequencing (NGS) technologies including PacBio RS II platform have been successfully utilized for genome sequencing of various pathogenic *Diaporthe* species, such as *D. citri* (*Liu et al., 2021b*), *D. phragmitis* (*Wang et al., 2021b*), *P. longicolla* (*Zhao et al., 2021*), *D. ilicicola* (*Emanuel et al., 2022*). Hybrid assemblies combining PacBio and Illumina sequencing technologies have been shown to enhance assembly contiguity and accuracy. In our current study, we successfully obtained a high-quality genome sequence of *D. eres* P3-1W using a combination of PacBio and Illumina sequencing approaches.

A total of 857 CAZymes of six classes were identified in *D. eres* P3-1W. Our results also revealed the presence of six CAZymes families in eight other *Diaporthe* species, suggesting that all CAZymes families may be common features of this genus. A previous research has shown that genes encoding pectinase (GH28), cellulase (GH3 and GH12) and hemicellulose (GH11 and GH43) in the GHs family are related to *Valsa Mali* infecting apple (*Silva et al., 2020*). In this study, we found that a diverse range of genes associated with cellulase (GH3, −5, −6, −7, −12, and −45), pectinase (GH28), hemicellulose (GH11 and GH43), xylanase (GH10, −11, and −30) and chitinase (GH18) in *D. eres* P3-1W. These findings suggested that *D. eres* P3-1W may have the ability to degrade these substrates as significant carbon source in their natural environment. Additionally, our results indicated that this fungus can cause fruit rot in other species except for 'Hongyang' kiwifruit, hinting at a potentially broad host range for *D. eres* P3-1W. Furthermore, four enzymes (PG, PME, β-Gal and Cx) exhibited varying activities during *D. eres* P3-1W infection on 'Hongyang' kiwifruit. However, further research is need to understand the roles of these enzymes in the pathogenicity of this fungus.

## CONCLUSIONS

This study presents the identification of *D. eres* P3-1W as a causal agent of postharvest rot of 'Hongyang' kiwifruit in China. This study observed varying activities of four CWDEs (Cx, β-Gal, PG and PME) upon infection with *D. eres* P3-1W. A comprehensive understanding of the CWDEs was conducted by sequencing the genome of *D. eres* P3-1W using Illumina and PacBio technology. Our results revealed that the genome size of *D. eres* P3-1W was 58,489,835 bp, with an N50 of 5,939,879 bp and a GC content of 50.7%. Among the 15,407 predicted PCGs, 857 CAZymes were identified and included 221 AAs, 374 GHs, 91 GTs, 39 PLs, 24CEs, and 108 CBMs in *D. eres* P3-1W genome. The detailed genome description along with enzyme assay results will be convenient to enhance our understanding of the mechanisms underlying the pathogenicity of *D. eres* P3-1W.

### Funding

This work was supported by the Guizhou Science and Technology Department, grant number QianKeHeJiChu-ZK[2022]530 and the Scientific Research (Cultivation) Project of Liupanshui Normal University, grant number LPSSY2023KJZDPY06. The funders had no role in study design, data collection and analysis, decision to publish, or preparation of the manuscript.

### Grant Disclosures

The following grant information was disclosed by the authors:
Guizhou Science and Technology Department: QianKeHeJiChu-ZK[2022]530.
Scientific Research (Cultivation) Project of Liupanshui Normal University: LPSSY2023KJZDPY06.

### Competing Interests

The authors declare that they have no competing interests.

### Author Contributions

- Li-Zhen Ling conceived and designed the experiments, analyzed the data, prepared figures and/or tables, authored or reviewed drafts of the article, and approved the final draft.
- Ling-Ling Chen performed the experiments, prepared figures and/or tables, and approved the final draft.
- Zhen-Zhen Liu performed the experiments, prepared figures and/or tables, and approved the final draft.
- Lan-Ying Luo performed the experiments, prepared figures and/or tables, and approved the final draft.
- Si-Han Tai performed the experiments, prepared figures and/or tables, and approved the final draft.

- Shu-Dong Zhang conceived and designed the experiments, analyzed the data, prepared figures and/or tables, authored or reviewed drafts of the article, and approved the final draft.

## Data Availability

The genome data is available at NCBI: PRJNA1073476, SAMN39831279 and SRR27880200 and SRR28059423.

The raw data are available in the Supplemental Files.

## Supplemental Information

Supplemental information for this article can be found online at http://dx.doi.org/10.7717/peerj.17715#supplemental-information.

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
