# Peer review of "Genome sequencing and CAZymes repertoire analysis of Diaporthe eres P3-1W causing postharvest fruit rot of ‘Hongyang’ kiwifruit in China"

_PeerJ, doi:10.7717/peerj.17715_

## Round 0.1 · original submission · Major Revisions

Your manuscript indicates the intent to go beyond a mere genome report by exploring mechanisms of pathogenicity in the postharvet rot fungus Diaporthe eres P3-1W. While this endeavour is of broad interest and suitable for the readership of PeerJ, I agree with the two expert reviewers that revision and additional analysis are required to support the biological conclusions drawn by the authors. Both reviewers have made very constructive and helpful suggestions to increase the biological insight and help shape this manuscript into a well-rounded story. Please also note that reviewer 1 has uploaded a PDF with more detailed comments and suggestions for you.

The most important points raised by the expert reviewers:

- More details on geographic extent of the dry rot, sample sizes, and overall proportion of Diaporthe isolates of all isolates, are neccessary for the fungal isolation and results sections.
- While stated at the end of the abstract, pathology and pathogenicity is not further explored in the manuscript.
- Both reviewers have pointed out that the phylogenetic relationships of the isolate characterized in the study would require further attention. Reviewer 2 suggested that exploring the CAZyme repertoire in greater depth would be insightful and would increase the robustness of the analyses and conclusions, and might provide some support to extend the findings to other species. Such additional analyses could include a broader selection of genomes of Diaporthe strains, species, and/or closely related species infecting similar hosts. Here, further statistical testing of differences in CAZyme counts across species along with multivariate analyses to characterize CAZyme distribution could be insightful.
- As pointed out by reviewer 1, the discussion would benefit from diving deeper into versatility of lifestyles and functional guilds of Diaporthe species, in particular in the context of the importance of CAZymes in a real-world scenario. As such, the manuscript would also benefit from an integration of the enzyme essays into the overall story.
- Finally, the manuscript could benefit from a more thorough exploration of how the results might influence future strategies for the management of soft rot disease.
- I would also like to encourage the authors to seek language editing services to improve the readability of the manuscript.

**Language Note:** The Academic Editor has identified that the English language must be improved. PeerJ can provide language editing services - please contact us at [email protected] for pricing (be sure to provide your manuscript number and title). Alternatively, you should make your own arrangements to improve the language quality and provide details in your response letter. – PeerJ Staff

·

Basic reporting

This manuscript describes the generation of a genome sequence from a Diaporthe isolate, collected from a rotten post-harvest fruit. There are a bunch of comments added to the manuscript draft.
Broad points:
• The writing of the manuscript requires tightening and oversight from an English speaker. There are problems with 1. Sections that need to be refocused for understanding; 2. Use of words or phrases that disrupt the context of sentences (it’s often possible to see the intent, but the reading is much harder work than it needs to be). See some specific examples on the submission.

Experimental design

The sequencing methodology seems competently performed and reasonably written, although some of the English is still idiosyncratic (less so than the Introduction). The methodology for the enzyme assays is noticeably more tightly written.

• More details are required on the methods to isolate the Diaporthe (e.g. l105-) and on the outcomes of the microbiology. How widespread is the Hongyang fruit rot? How many samples were looked at? How many fungi were isolated/identified/how many were of this Diaporthe species? This, of course, would need to be reflected in the Results too. At present I don't see a satisfactory exploration of pathology in the manuscript.

Validity of the findings

• Probably my main suggestion about this work is that it needs either to be a short genome report (as it effectively is now), or there needs to be more explanation and possibly work, to bring in some biology. At present the work is effectively a genome assembly and resulting gene list.
• The phylogenetics of this isolate is given insufficient depth of coverage (l226-). That this is a matter of interest is shown by the list of other Diaporthe (l324-) recorded from kiwifruit. However, their phylogenetic relationship to this isolate needs to be better explored. The choice of citations here should also be improved or enlarged. I’m at least familiar with long established research on the ubiquity and identity of Diaporthe in New Zealand kiwifruit orchards (Manning… Manning et al 2003…Hawthorne et al 1982). The manuscript implies that this is the first record of D. eres from Hongyang and/or the occurrences of D. eres on kiwifruit could be better discussed. I notice there are multiple D. eres from kiwifruit listed in the New Zealand ICMP collection, including from Hongyang.
• Diaporthe are saprophytes, endophytes, pathogens. Diaporthe are involved in leaf spotting as well as postharvest fruit rots in orchards. An ability to grow in a ripening fruit (especially a damaged one) is a particular type of “pathogenicity”. The discussion barely mentions this versatility or how the CAZymes might work in the real, wider, orchard world.
• The authors have explored the activity of four enzymes. I imagine that this was an attempt to link the genome sequence to biology. However, I don’t think the enzyme assays really effectively contribute to this story at the moment.
• The selection of citations in places is odd – see start of the Discussion for example.

Additional comments

see previous box?

·

Basic reporting

The manuscript would benefit significantly from a comprehensive review of the English language usage. Certain sentences require grammatical corrections, particularly with the use of definite articles and connectives. Additionally, the text contains overly complex constructions and could be simplified to enhance clarity. The frequent use of passive voice and lengthy descriptions could be revised for conciseness. I recommend a thorough language revision to ensure that the style of writing complements the importance of the findings and facilitates reader comprehension.

The discussion section of the paper could benefit from a more thorough exploration of how the results might influence future strategies for managing soft rot disease; some recent papers even suggest treatment options targeting this same pathogen (Tian et al, 2023: https://doi.org/10.3390/foods12173197; Wang et al, 2022: https://doi.org/10.3390/horticulturae8070624).

The labeling for functions in Figure S2 are a bit hard to read; changing them from grey to black or increasing the font size is recommended.

Experimental design

The manuscript's choice to analyze the CAZyme repertoire in Diaporthe eres compared to other genomes is intriguing but could be expanded for greater depth. Specifically, the rationale for selecting these species—pathogens of diverse crops like soybean, citrus, and grapevine—raises questions about their relevance to the study's focus on a kiwifruit pathogen. Including a broader array of genomes, particularly additional strains or closely related species affecting similar hosts, would enhance the robustness of the analysis. This expansion would not only provide a more comprehensive view of CAZyme diversity but also clarify the evolutionary adaptations across species with different host specificities. The inclusion of multiple genomes from the same species or closely related species would strengthen the study by offering insights into intra-species variability and the evolutionary pressures shaping these enzymes.

Validity of the findings

The CAZyme analysis in the manuscript, while methodologically sound in gene annotation and prediction using dbCAN, lacks robust statistical analysis, which is crucial for validating the comparative insights presented. Specifically, the study would benefit from incorporating statistical tests such as ANOVA to substantiate the differences in CAZyme counts across species. Additionally, multivariate analyses like PCA and cluster analysis could reveal deeper patterns in CAZyme distribution that correlate with pathogenic traits or host specificities. Including phylogenetic comparative methods would also enhance the interpretation by accounting for evolutionary relationships among species. These statistical enhancements are essential for strengthening the conclusions about the evolutionary adaptations and functional implications of CAZymes in pathogen-host interactions.

The presentation and analysis underlying Figure 5 and Table S4 lack clarity and detail. It is assumed that the asterisks (*) in Figure 5 denote statistical significance; however, this is not explicitly stated in the figure legends or accompanying text. Furthermore, if these symbols are indeed indicators of statistical significance, the manuscript fails to describe the statistical methods employed to determine this. For a comprehensive understanding and validation of the results, it is essential to clearly specify the meaning of such annotations and thoroughly describe the statistical tests used.

The conclusions link the genomic and enzymatic characteristics of Diaporthe eres P3-1W to its ability to cause postharvest rot in 'Hongyang' kiwifruit. While the study effectively supports this link for D. eres P3-1W, it cautiously hints at broader implications for similar species. However, extending these findings to other species or conditions without additional evidence may be premature.

---

## Round 0.2 · accepted · Accept

Thank you for the thorough revision of your manuscript. The expert reviewer is satisfied with your revision, and I wholeheartedly agree with them. The manuscript is now acceptable to be accepted for publication in PeerJ.

·

Basic reporting

In previous comments, a thorough revision of the English language usage was recommended, as well as text conciseness and clarity. These remarks were addressed and corrected.

Some suggestions on references and context were made, and denied, but the author's explanation on why they chose to do so was sufficient.

Experimental design

The critics on experimental design (on the genomes chosen for the comparative genomics analyzes) were addressed. The author did not make the changes advised, but explained the rationale behind their choices, in this case, the lack of deposited genomes.

Validity of the findings

All comments regarding statistical analyzes and methods clarification were properly addressed.